# Crustal rejuvenation stabilised Earth's first cratons

Jacob A. Mulder [1] ✉, Oliver Nebel[1], Nicholas J. Gardiner[1,2], Peter A. Cawood [1], Ashlea N. Wainwright [3] & Timothy J. Ivanic[4]

The formation of stable, evolved (silica-rich) crust was essential in constructing Earth's first cratons, the ancient nuclei of continents. Eoarchaean (4000–3600 million years ago, Ma) evolved crust occurs on most continents, yet evidence for older, Hadean evolved crust is mostly limited to rare Hadean zircons recycled into younger rocks. Resolving why the preserved volume of evolved crust increased in the Eoarchaean is key to understanding how the first cratons stabilised. Here we report new zircon uranium-lead and hafnium isotope data from the Yilgarn Craton, Australia, which provides an extensive record of Hadean–Eoarchaean evolved magmatism. These data reveal that the first stable, evolved rocks in the Yilgarn Craton formed during an influx of juvenile (recently extracted from the mantle) magmatic source material into the craton. The concurrent shift to juvenile sources and onset of crustal preservation links craton stabilisation to the accumulation of enduring rafts of buoyant, melt-depleted mantle.

[1] School of Earth, Atmosphere and Environment, Monash University, Clayton, VIC, Australia. [2] School of Earth and Environmental Sciences, University of St. Andrews, St. Andrews, United Kingdom. [3] School of Geography, Earth and Atmospheric Sciences, University of Melbourne, Parkville, VIC, Australia. [4] Geological Survey of Western Australia, Department of Mines, Industry Regulation and Safety, East Perth, WA, Australia. ✉email: jack.mulder@monash.edu

The transition between the Hadean and Eoarchaean (approximately 4000 Ma) marks the time when evolved crustal rocks—the essential building material of continents—started to become widely preserved on Earth. The delayed arrival of this stable crust, over 500 Ma after planetary accretion, has traditionally been attributed to the wholesale destruction of pre-existing Hadean crust either by intense meteorite bombardment[1], mantle overturns[2], or subduction[3]. However, these models continue to be challenged by discoveries of intact remnants of Hadean crust[4,5], recycled Hadean zircons in younger rocks[6–8], and Hadean radiogenic Pb and $^{142}$Nd isotopic signatures in several cratons worldwide[5,9–11], which indicate that potentially large tracts of Hadean crust survived the Hadean–Eoarchaean transition. An alternative view suggests that the Hadean Earth was surfaced by a predominantly mafic-ultramafic crust, and the Eoarchaean witnessed a fundamental change in crust-forming processes that produced the first widespread stable, evolved crust[10,12–14]. Testing this hypothesis has, however, proved challenging due to the scarcity and geological complexity of the ancient terranes that preserve a record of crust formation spanning the Hadean and Eoarchaean.

Preserved evolved Eoarchaean crust is dominated by rocks of the tonalite–trondjhemite–granodiorite (TTG) suite. TTGs mostly form by a two-stage process: initial extraction of primary mafic crust through partial-melting of the mantle, followed by later partial melting of that mafic crust to form evolved magma[15]. The mafic precursors of TTGs may be either: (i) juvenile reservoirs, which are remelted shortly after being extracted from the mantle and emplaced in the crust; or (ii) ancient reservoirs that have endured in the crust for hundreds of million years. The $^{176}$Lu-$^{176}$Hf decay system is well-suited to characterising the source of TTGs, as the fractionation of Lu from Hf during partial melting results in the mantle (high $^{176}$Lu/$^{177}$Hf) and crust (low $^{176}$Lu/$^{177}$Hf) developing distinct time-integrated radiogenic Hf isotopic compositions. Consequently, Eoarchaean TTGs sourced from juvenile reservoirs inherited the near-chondritic Hf isotopic composition of the Eoarchaean mantle[16,17], whereas Eoarchaean TTGs sourced from ancient reservoirs acquired less radiogenic (sub-chondritic) Hf isotopic signatures. Zircon is an ideal tool to study the Hf isotopic character of ancient TTGs as it is a ubiquitous phase in evolved magmas and preserves the Hf isotope signature of its parental melt owing to its low Lu/Hf (<0.001) and resilience to isotopic resetting. Coupling in situ U–Pb dating and Hf isotope analysis of zircon provides a time-integrated record of the source characteristics of TTGs, thus offering key insights into evolved crust-forming processes on the early Earth.

Here, we focus on the Yilgarn Craton, Western Australia (Fig. 1), which contains exposures of some of the oldest known evolved Eoarchaean crust[18] and preserves Earth's most complete temporal archive of Hadean evolved magmatism through 4370–4000 Ma detrital zircons[6–8,14]. The Yilgarn Craton therefore provides an important opportunity to examine the nature of evolved magmatism across the critical Hadean–Eoarchaean interval over which the majority of Earth's first stable, evolved crust formed. New zircon U–Pb and Hf isotopic data reveal a link between the onset of crustal preservation in the Yilgran Craton and a fundamental shift in the source characteristics of evolved magmatism across the Hadean–Eoarchaean transition.

## Results

**Hadean–Archaean zircon Hf isotope record of the Yilgarn Craton.** The earliest crustal record of the Yilgarn Craton is recorded by Hadean and early Eoarchaean (>3800 Ma) detrital zircons, the majority of which are found in approximately 3000 Ma sedimentary successions in four locations (Fig. 1). The most intensely studied locations include exposures of metamorphosed

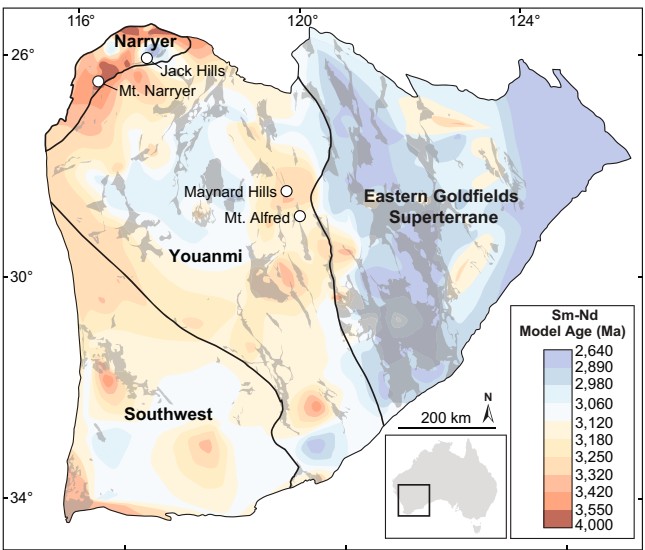

**Fig. 1 Tectonic terrane map of the Yilgarn Craton, Western Australia.** Main tectonic terranes are labelled and separated by thick black lines. Greenstone belts are shown in dark grey. Contoured Sm-Nd model ages from whole rock analyses are from ref. [21]. and highlight the distribution of ancient crust exposed in the Narryer Terrane and its possible continuation beneath younger sequences in the western part of the craton. White circles mark the location of ~3000 Ma sedimentary sequences containing Hadean and early Eoarchaean detrital zircons.

sandstone and conglomerate in the Jack Hills and at Mount Narryer, both within the Narryer Terrane[3,6–8,19] (Fig. 1). The other two locations occur approximately 400 km to the southeast, in the Youanmi Terrane, and include exposures of metamorphosed sandstones of the Illaara Formation in the Maynard Hills and at Mount Alfred[20,21] (Fig. 1). We complement previously published data from the Jack Hills and Mount Narryer[3,14] with a large new U–Pb–Hf isotopic dataset from detrital zircons ($n = 2296$) from the Illaara Formation. A subset of 638 detrital zircons showing the least evidence for disturbance of their U–Pb and Hf isotope systematics are interpreted to best represent the age and Hf isotopic composition of their evolved magmatic source rocks (Supplementary Data 1). Our new zircon ages range from 4150 to 3250 Ma and form broad age populations with peaks at approximately 3750, 3650, 3500, and 3400 Ma (Fig. 2; Supplementary Fig. 1). These age populations are similar to those observed in detrital zircons of the Jack Hills and Mount Narryer metasedimentary rocks and overlap temporally with early Archaean crust exposed in the Narryer Terrane (Fig. 2; Supplementary Fig. 1). Therefore, the ~3000 Ma sedimentary successions in the Yilgarn Craton are interpreted to provide a broad sampling of the Narryer Terrane and related early Archaean crust that may underlie much of the western Yilgarn Craton[21] (Fig. 1).

Detrital zircons from the Illaara Formation with ages older than 3800 Ma are uncommon (~1.5% of analyses) and yield subchondritic Hf isotopic compositions. These unradiogenic Hf isotopic compositions equate to negative $\varepsilon Hf_{(t)}$ values (the deviation in $^{176}$Hf/$^{177}$Hf from the chondritic uniform reservoir in parts per 10,000 at the time of zircon crystallisation) and support previous studies of the least disturbed Jack Hills detrital zircons[14] that suggest most of the earliest evolved magmas in the Yilgarn Craton were not derived from juvenile precursors. Instead, the least disturbed Hadean Jack Hills zircons and the majority of pre-3800 Ma zircons from the Illaara Formation collectively define an approximately linear $\varepsilon Hf_{(t)}$–time array initiating at $\varepsilon Hf_{(t)} = -1.5$ at 4400 Ma and evolving to $\varepsilon Hf_{(t)} = -3$

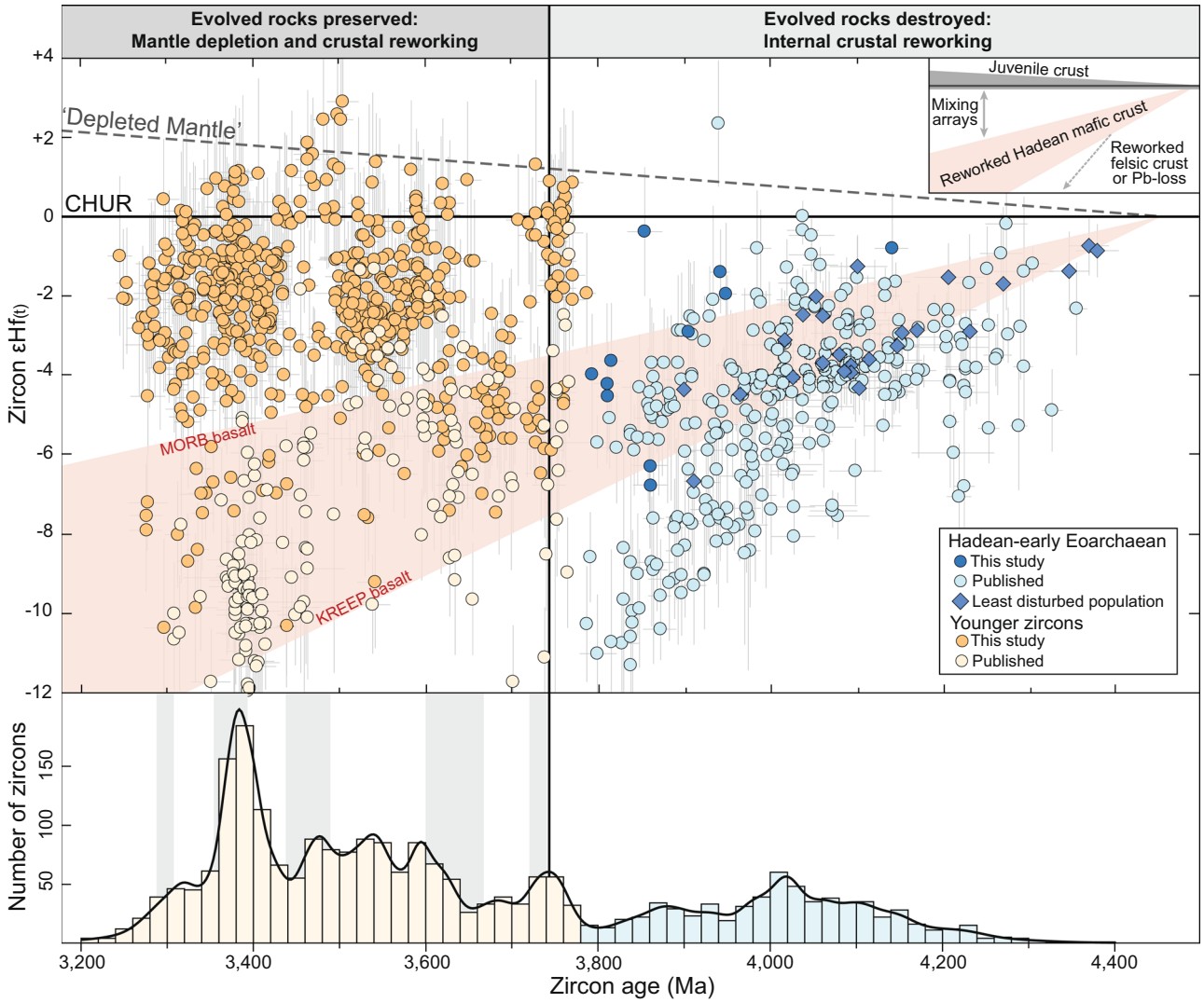

**Fig. 2 Plot of initial epsilon hafnium (εHf$_{(t)}$) versus zircon age for Hadean and Archaean detrital zircons from the Yilgarn Craton.** Most pre-3800 Ma detrital zircons define a Hf isotopic array consistent with internal reworking of early Hadean mafic sources such as typical mid ocean ridge basalt ($^{176}$Hf/$^{177}$Hf = 0.026) or Hadean incompatible element enriched lunar basalt (KREEP, $^{176}$Lu/$^{177}$Hf = 0.020; ref. [14].). From ~3750 Ma onwards, detrital zircon age populations define vertical εHf$_{(t)}$–time groups that reflect mixing between juvenile sources and pre-existing Hadean crust. The grey dashed line defining the depleted mantle is approximately fitted through the most juvenile zircon analyses. Error bars are 2SE. CHUR chondritic uniform reservoir. Only published data collected by concurrent Pb–Hf analysis are shown[3,14]. A kernel density estimate curve and histogram of zircon ages (new and previously published) are shown at the bottom of the diagram. The grey bars behind the histogram correspond to the age of crust preserved in the Narryer Terrane[18].

to −7 by 3800 Ma (Fig. 2). This Hf isotopic array is consistent with formation of the parental magmas of the pre-3800 Ma zircons through prolonged (~600 Ma) internal reworking of an older, predominantly mafic crust ($^{176}$Lu/$^{177}$Hf = 0.020–0.026) that was extracted from the mantle in the early Hadean[14,19].

In contrast to the Hadean–early Eoarchaean record, our new zircon data document a step-change in the Hf isotopic evolution of the Yilgarn Craton at ~3750 Ma (Fig. 2). From this time onwards, the zircon populations define vertical groups in εHf$_{(t)}$–time space that are anchored between an approximately chondritic endmember (εHf$_{(t)}$ = 0 to +1) and the extrapolation of the Hadean mafic crustal reworking trend defined by the pre-3800 Ma zircons. We interpret each of these vertical εHf$_{(t)}$–time groups to result from the initial extraction of juvenile mafic magma from the mantle, its emplacement into the Hadean mafic crustal substrate sampled by the pre-3800 Ma zircons, followed by rapid reworking of both the juvenile and Hadean mafic crustal reservoirs to produce evolved magmas (Fig. 2). This multi-source

and multi-stage evolution contrasts with the comparatively simple reworking trend defined by the older (>3800 Ma) zircons and signals a fundamental change in the source characteristics of zircon-bearing magmas in the Yilgarn Craton[3,13,14].

## Discussion

The critical observation revealed by our new zircon-Hf isotope data is that the onset of crustal preservation in the Yilgarn Craton at ~3750 Ma[18] coincides with the step-change in magmatic source characteristics recorded by detrital zircons (Fig. 2). This finding may implicate a link between crustal rejuvenation—defined here as an influx of isotopically juvenile magma into an ancient crustal substrate—and the stabilisation of evolved Eoarchaean crust. Indeed, a temporal link between a step-change in the Hf isotope evolution and the appearance of the oldest preserved evolved crustal rocks is also evident in several other cratons with well-documented Hadean–early Eoarchaean zircon archives (Fig. 3a, b). Detrital and xenocrystic Hadean–Eoarchaean zircons in the

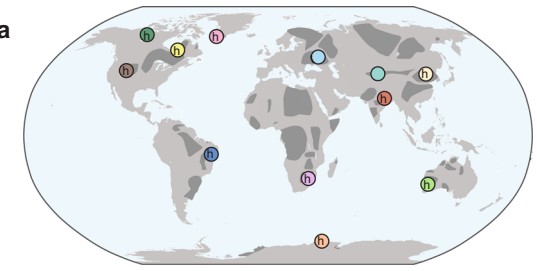

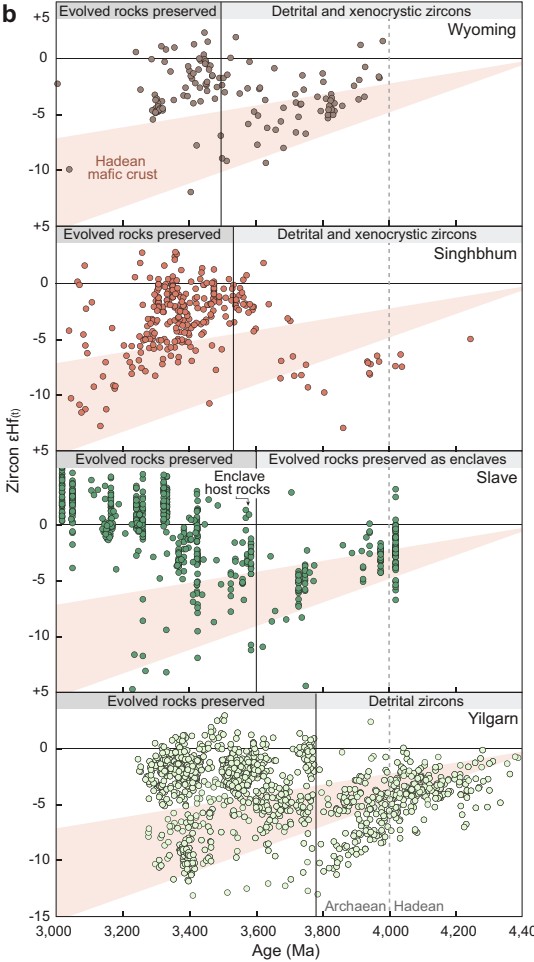

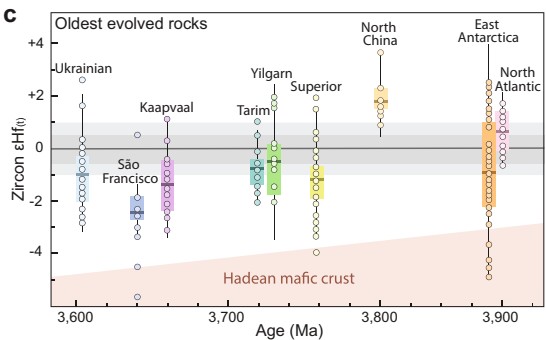

**Fig. 3 Zircon Hf isotopic systematics of Earth's oldest evolved rocks.**
**a** World map showing Archaean cratons in dark grey (modified from ref. [23]). Coloured circles show Eoarchaean cratons with data plotted in panels **b** and **c**. Eoarchaean cratons preserving Hadean zircons, radiogenic Pb, or [142]Nd isotopic signatures are labelled 'h'. **b** Zircon $\varepsilon Hf_{(t)}$–time plots for Archaean cratons that record a link between a step-shift in Hf isotope evolution and the preservation of evolved rocks. Black vertical line marks the age of the oldest rock dated by U–Pb zircon geochronology. Dashed grey vertical line shows the Hadean–Eoarchaean boundary. The Hf isotopic evolution of the Hadean mafic crustal reservoir documented in the Yilgarn Craton is shown in red. **c** Box plots of zircon $\varepsilon Hf_{(t)}$ data from the oldest TTGs in different Eoarchaean cratons. Box limits define upper and lower quartiles, the thick horizontal line shows the mean, whiskers show 1.5 x interquartile range, points show individual analyses. Horizontal grey bars show chondritic reference values ±0.5 and ±1 εHf units. See Supplementary Note 1 for sources and details of compiled data.

The emplacement of the voluminous ~3600 Ma granitoids hosting these enclaves was synchronous with a pronounced and sustained shift in the Hf isotopic evolution of the Slave Craton to more juvenile sources[22] (Fig. 3b), indicating a possible causative relationship between crustal rejuvenation and an increase in the volume of stable evolved crust.

Although lacking extensive Hadean zircon archives, the preservation of Hadean Pb and [142]Nd isotopic signatures in many other Eoarchaean cratons[23] suggests they were built on substrates of Hadean crust[9–11] (Fig. 3a). Strikingly, the oldest evolved rocks preserved in most Eoarchaean cratons have near-chondritic mean zircon Hf isotopic compositions that require an important contribution from juvenile precursors in their formation (Fig. 3c, ref. [17]). However, the spread from suprachondritic (positive $\varepsilon Hf_{(t)}$) to less radiogenic (negative $\varepsilon Hf_{(t)}$) endmembers implies that these TTGs were not purely juvenile additions to the crust, but were emplaced within, and extensively reworked their Hadean crustal substrates[24]. In summary, the Hf isotope data from other ancient terranes are consistent with the more temporally complete archive of Hadean–Eoarchaean evolved magmatism preserved in the Yilgarn Craton and support a temporal link between crustal rejuvenation and the stabilisation of the Earth's oldest cratons in the Eoarchaean.

The temporal link between crustal rejuvenation and stabilisation is readily explained by the formation of lithospheric mantle beneath nascent Eoarchaean cratons. The removal of iron and volatile components during partial melting of the mantle to form juvenile magma produces a buoyant and rigid residue of melt-depleted lithospheric mantle[25,26]. The enhanced mechanical strength and buoyancy of this melt-depleted residue stabilises it against erosion by the convecting mantle. Roots of strongly depleted lithospheric mantle, locally >200 km thick, are found beneath most Archaean cratons and are widely interpreted to have facilitated the long-term preservation of the overlying crust[25–28]. This empirical evidence for a link between crustal stabilisation and the development of lithospheric mantle is supported by numerical models that demonstrate that even with a more vigorously convecting mantle on a hotter early Earth, the presence of lithospheric mantle substantially increases the preservation potential of crust[29,30]. Dating of the lithospheric mantle beneath Archaean cratons indicates that it mostly formed synchronously with the overlying crust, reinforcing the coupled development of stable crust and lithospheric mantle[27,28]. Importantly, lithospheric mantle xenoliths from several cratons with Re–Os mantle model ages in excess of 3600 Ma[27] and rare exposures of ~3800 Ma melt-depleted mantle rocks[31] provide independent evidence for the development of lithospheric mantle

Wyoming and Singhbhum cratons record a prolonged period of reworking of Hadean crust prior to ~3600 Ma, followed by a shift to more juvenile sources approximately coincident with the appearance of stable TTGs (Fig. 3b). Internal reworking of Hadean crust was also important in producing the earliest evolved crust in the Slave Craton, which is preserved as decimetre-scale enclaves in the Acasta Gneiss Complex, Canada[4].

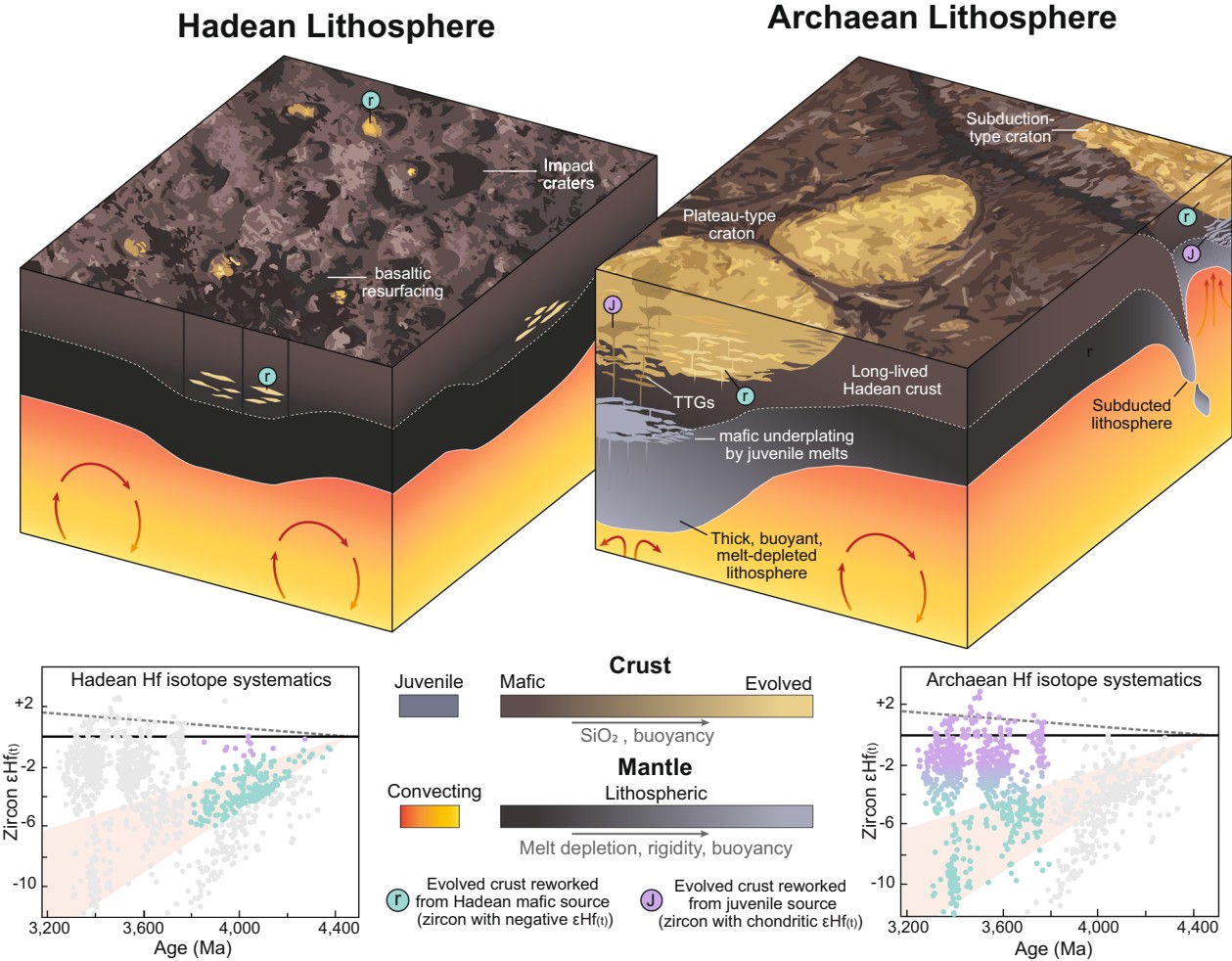

**Fig. 4 Schematic diagrams illustrating changes in evolved crust-forming processes and lithospheric architecture between the Hadean and Archaean.**
The Hadean crust is predominantly mafic–ultramafic with minor evolved melts formed by internal differentiation or impact melting. Hadean crust persists into the Archaean and is reworked in subduction zones or thick crustal plateaus above mantle upwellings to form evolved magmas that crystallise zircon with negative $\varepsilon Hf_{(t)}$. These two tectonic settings also facilitate adiabatic or flux melting of convecting mantle to form juvenile mafic magmas, which are reworked to produce evolved magmas that crystallise zircon with approximately chondritic $\varepsilon Hf_{(t)}$. Juvenile melt extraction produces a complementary melt-depleted, rigid, and buoyant lithospheric mantle, which stabilises the evolved crustal nuclei of Earth's first cratons. The zircon $\varepsilon Hf_{(t)}$–time plots show the corresponding zircon-Hf isotope characteristics of evolved crust from the Yilgarn craton between the Hadean and Archaean eons.

contemporaneously with the appearance of the first stable, evolved crust in the Eoarchaean.

Although the development of stabilising depleted lithospheric mantle provides a plausible explanation for the increase in crustal preservation across the Hadean–Eoarchaean transition, the geodynamic trigger for this possible change in lithospheric architecture remains speculative. The scarcity of zircons with juvenile Hf isotopic signatures in the Yilgarn Craton prior to ~3800 Ma is compatible with stagnant-lid geodynamic models for the Hadean in which low-volume, evolved melts formed through remelting of mafic–ultramafic crust below overthickened volcanic centres[10,14] or by meteorite impacts[32] (Fig. 4). The switch to reworking of both Hadean and juvenile sources now recognised in the zircon Hf isotopic record of most early Archaean cratons between ~3850 and 3600 Ma (Fig. 3b, c) marks a fundamental and sustained change in the geodynamic setting of evolved crust production[13]. The observation that different cratons stabilised at different times over a period of at least 250 Ma (Fig. 3c) suggests the change in global geodynamic regime was a transitional process[13,33], rather than a response to a single cataclysmic event[2]. Previous studies have noted the similarity of these zircon-Hf isotope–time arrays to those produced in modern convergent margin systems[34,35] and

suggested that the global step-shifts in zircon-Hf isotope systematics (Fig. 3b) reflect the progressive onset of mobile-lid tectonics[3,13,22]. However, the extraction of isotopically juvenile precursors to zircon-bearing magmas is not unique to convergent margin settings. Early Archaean plateau-type terranes, such as the archetypal East Pilbara Terrane[12,36], are inferred to have formed in a poorly mobile lid setting above long-lived mantle upwellings[37,38], which presents an equally viable geodynamic setting for the production of evolved magmas with juvenile zircon-Hf isotopic compositions[17,39]. Although zircon-Hf isotope data alone is not diagnostic of the specific geodynamic processes operating in early Archaean, the key advance made by this study is to highlight that the distinct change in the geodynamic setting of crust formation across Hadean–Archaean transition was coincident with the stabilisation of the first cratons.

A link between craton stabilisation and a change in global geodynamics is supported by recent numerical models describing the thermochemical differentiation of the lithosphere across the Hadean–Archaean transition[40]. This modelling predicts that following long-term (~500 Ma) stretching, a lithospheric lid will begin to thin and segment, resulting in large-scale decompression melting of the underlying convecting mantle. Due to the negative

feedback between strain migration and lithospheric stiffening that accompanies melt-extraction, rift migration incorporates large volumes of rigid, melt-depleted mantle into the lithosphere, triggering the stabilisation of cratons. This process develops distinct geodynamic settings for crust formation depending on the strength of the modelled lithosphere. For a high strength lithosphere, new crust is produced over broad mantle upwellings below thinned regions of a poorly mobile lid, in a setting analogous to early Archaean plateau-type terranes. In contrast, mobile-lid behaviour can develop under lower lithospheric strengths as rifting of the lid into discrete tectonic plates is compensated by the development of subduction-like lithospheric downwellings[30,40]. In the context of this modelling, our new findings provide important geochemical and geological evidence for a fundamental change in the geodynamic setting of crust formation during the Hadean–Archaean transition, which facilitated the extraction of juvenile melts, crustal reworking, evolved magmatism, and the production of stabilising melt-depleted lithospheric mantle, which were critical to forming Earth's first cratons.

## Methods

**U–Pb–Hf isotopic analysis of zircon.** Zircons were isolated from approximately 5 kg samples using standard magnetic and heavy liquid separation techniques. Zircons were cast into 25 mm round epoxy mounts, polished, and imaged using a Gatan PanaCL panchromatic CL detector housed at the University of Melbourne, Australia to characterise internal textures and aid in analytical spot placement.

U–Pb and Hf isotopic data were collected via laser ablation split-stream inductively coupled plasma mass spectrometry at Monash University, Australia. The analytical set-up includes an ASI RESOLution 193 nm laser ablation system coupled to a Thermo Fisher iCAP TQ Triple Quadrupole inductively coupled plasma mass spectrometer for measurement of U and Pb isotopes and a Thermo Scientific Neptune Plus multicollector inductively coupled plasma mass spectrometer for measurement of Lu, Yb, and Hf isotopes. The ablated sample aerosol was split evenly using a Y-connector and transported to the mass spectrometers using polyurethane tubing with $N_2$ gas added to the aerosol stream prior to arriving at the torch to enhance sensitivity. Laser ablation was preformed using a 35 μm spot, a frequency of 8 Hz, and a fluence of approximately 4.5 J/cm$^2$ (measured with an external fluence meter). Hafnium isotopes were collected following the method outlined in ref. [41], which involved 30 s of background measurement and 60 s of ablation with a 1 s on-peak integration time. Dwell times for U, Th, and Pb isotopes on the iCAP TQ were 10 ms for $^{238}U$ and $^{232}Th$, 20 ms for $^{208}Pb$, 70 ms for $^{207}Pb$, 40 ms for $^{206}Pb$ and 30 ms for $^{204}Pb$ and $^{202}Hg$. Isotopic data were reduced with the Iolite 3 software package[42]. The U–Pb data was reduced with the U_Pb_Geochron4 data reduction scheme of Iolite with a smooth cubic spline used to model down-hole fractionation. The Hf isotopes were reduced using the Hf_isotopes data reduction scheme of Iolite with Yb mass bias corrected assuming $^{173}Yb/^{171}Yb = 1.132685$ (ref. [43]) and Hf mass bias corrected assuming $^{179}Hf/^{177}Hf = 0.7325$ (ref. [44]).

Natural zircon reference materials were measured every 15 unknowns during each analytical run and the drift between reference material brackets was corrected using a spline function. OG1 zircon[45] was used to calibrate the U–Pb data and the Mud Tank zircon[46] was used to calibrate the Hf isotope data. Secondary standards were used to validate U–Pb and Hf results and included the reference zircons 91500[47], GJ1[48], Plešovice[49], and QGNG[50]. The differences in the down-hole U–Pb fractionation behaviour of reference zircons with significantly different ages was accounted for by calibrating the U–Pb data for QGNG (1851 Ma) against OG1 (3465 Ma) and calibrating the U–Pb data for the younger reference zircons Plesovice (337 Ma), GJ1 (600 Ma), and Mud Tank (732 Ma) against 91500 (1063 Ma). The weighted mean U–Pb ages and $^{176}Hf/^{177}Hf$ measured for the reference zircons agree with the published recommended values[45–54], as do the stable Hf isotope ratios measured over all analytical runs[55,56] (Supplementary Figs. 3 and 4). The ages reported for unknowns are $^{207}Pb/^{206}Pb$ ages with 2SE propagated uncertainties calculated by iolite and are uncorrected for common Pb. Only analyses with <±10% discordance between their $^{206}Pb/^{238}U$ and $^{207}Pb/^{206}Pb$ ages are considered in our interpretations. The $^{176}Hf/^{177}Hf$ and epsilon Hf values (eHf$_{(t)}$) at the time of crystallisation of the unknown zircons are calculated using their $^{207}Pb/^{206}Pb$ ages, the $^{176}Lu$ decay constant of ref. [57], and the CHUR parameters of ref. [58]. The kernel density estimation and histogram of zircon U–Pb ages and all weighted mean calculations were made using isoplotR[59].

## Data availability

All data used in this manuscript are included in the Supplementary Information File and Supplementary Data File 1.

## Code availability

The Iolite code used for U–Pb and Hf isotope data reduction is available at https://iolite.xyz/.

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

## Acknowledgements

This work was funded by Australian Research Council grant FL160100168 and Australian Research Council grant DP180100580. R. Pierson, M. Raveggi, and G. Hutchinson are thanked for technical assistance. T.J.I. publishes with permission from the Executive Director of the Geological Survey of Western Australia.

## Author contributions

J.A.M and O.N conceived the project, conducted the field work, and collected the isotope data. P.A.C. and N.J.G assisted with geological interpretations and data compilation. A.N.W. assisted in collecting the isotope data. T.J.I. co-ordinated and assisted with the field work. All authors analysed the data and contributed to writing the paper.

## Competing interests

The authors declare no competing interests.
