## [Peer Review File · Nature Communications]

REVIEWERS' COMMENTS

Reviewer #1 (Remarks to the Author):

Review of "Crustal rejuvenation stabilised Earth's first cratons" by Mulder et al.

This study presents new U-Pb and Hf isotope data for Hadean–Eoarchean detrital zircons from the Illaara Formation of the Yilgarn craton, Western Australia. The data highlight a pronounced shift in the Hf isotope–U–Pb age pattern at ~3.75 Ga: zircons older than ~3.75 Ga crystallized from magmas formed by remelting of Hadean crust, whereas younger ones record the additions of mantle-derived material. The timing of this isotopic shift is coincident with the crystallization ages of the oldest preserved rocks in the craton. Considering that partial melting of the mantle leaves residual mantle that is depleted in Fe and volatiles and therefore buoyant and stiff, the authors interpret the coincidence as evidence that the emergence of stable cratons resulted from significant juvenile magmatism.

I reviewed a previous version of this manuscript submitted to Nature Geoscience. My original review was critical of the novelty of the manuscript. The model that Archean cratons were stabilized due to melt-extraction from the mantle is long-held, as stated in the present manuscript (l. 25–27). In addition, the ~3.75 Ga isotopic shift in the Hadean–Eoarchean zircon record was previously appreciated and its geologic significance has been a subject of debate (Bell et al., 2014 GCA; Bauer et al., 2020 GPL). As the authors claim in the text and response to the comments, this study is certainly new in that the fundamental isotopic shift in the zircon record is temporally linked with the onset of the crustal preservation. However, I am not convinced that the temporal coincidence provides evidence for the craton stabilization by the significant melt extraction from the mantle because there are other possible mechanisms causing the zircon isotopic shift such as the onset of mobile-lid tectonics (Bell et al., 2014; Bauer et al., 2020). I would leave to the editor the decision whether linking the long-held model with the previously-recognized zircon isotopic shift whose geologic significance is controversial is worthy of publication in Nature Communication. Otherwise, this paper contains high quality data and is well written.

Minor comments:

l. 58: The ^{176}Lu – ^{177}Hf decay >> Th ^{176}Lu – ^{176}Hf decay.

l. 71–118: It should be acknowledged in this section that some studies previously highlighted the fundamental Hf isotopic shift at ~3.8 Ga in the Hadean–Archean detrital zircon record (e.g., Bell et al., 2014; Bauer et al., 2020). Although these studies are included in the reference list, the main text gives an impression that the newly obtained data in this study reveal the isotopic shift for the first time, which is not the case.

Reviewer #2 (Remarks to the Author):

'Crustal rejuvenation stabilised Earth's first cratons' by J. Mulder and colleagues

Upon reading this manuscript, I would reiterate my comments made on a previous version: it is clearly written, well illustrated, scientifically sound, and I believe would be of interest to the journal readership. The link between crustal stabilisation and formation of melt-depleted lithospheric mantle has been better explained and justified, and there is now greater attention to the geodynamic drivers, considering recent and relevant thermochemical/mechanical models. I am fully supportive of the publication of this manuscript and do not see how it could be easily improved within the scope of the study and the space restrictions.

A few minor points.

Line 26. A typo in the spelling of buoyant. Also in the caption to figure 4 and on the figure itself.

Line 55. I don't think ref. 55 is the most appropriate reference here. In my view, it is better to cite the seminal earlier papers that conceived this process – such as Barker and Arth (1976) *Geology*, or Rapp and Watson (1995) *J. Petrol.*

Line 191. Here it would be appropriate to cite Petersson et al 2020 (*Chemical Geology*), who established a CHUR-like zircon Hf isotopic composition for the oldest gneissic rocks of the Pilbara Craton. This study also provides support for the points made in lines 61–63 and 141–143.

Best regards, Tony Kemp, April 2021.

School of Earth, Atmosphere & Environment
9 Rainforest Walk, Clayton, Victoria, 3800, Australia
+61 3 9905 1595 Telephone
Jack.Mulder@Monash.edu Email

Reviewer #1 (Remarks to the Author):

Review of “Crustal rejuvenation stabilised Earth’s first cratons” by Mulder et al.

This study presents new U-Pb and Hf isotope data for Hadean–Eoarchean detrital zircons from the Illaara Formation of the Yilgarn craton, Western Australia. The data highlight a pronounced shift in the Hf isotope–U–Pb age pattern at ~3.75 Ga: zircons older than ~3.75 Ga crystallized from magmas formed by remelting of Hadean crust, whereas younger ones record the additions of mantle-derived material. The timing of this isotopic shift is coincident with the crystallization ages of the oldest preserved rocks in the craton. Considering that partial melting of the mantle leaves residual mantle that is depleted in Fe and volatiles and therefore buoyant and stiff, the authors interpret the coincidence as evidence that the emergence of stable cratons resulted from significant juvenile magmatism.

I reviewed a previous version of this manuscript submitted to Nature Geoscience. My original review was critical of the novelty of the manuscript. The model that Archean cratons were stabilized due to melt-extraction from the mantle is long-held, as stated in the present manuscript (l. 25–27). In addition, the ~3.75 Ga isotopic shift in the Hadean-Eoarchean zircon record was previously appreciated and its geologic significance has been a subject of debate (Bell et al., 2014 GCA; Bauer et al., 2020 GPL). As the authors claim in the text and response to the comments, this study is certainly new in that the fundamental isotopic shift in the zircon record is temporally linked with the onset of the crustal preservation. However, I am not convinced that the temporal coincidence provides evidence for the craton stabilization by the significant melt extraction from the mantle because there are other possible mechanisms causing the zircon isotopic shift such as the onset of mobile-lid tectonics (Bell et al., 2014; Bauer et al., 2020). I would leave to the editor the decision whether linking the long-held model with the previously-recognized zircon isotopic shift whose geologic significance is controversial is worthy of publication in Nature Communication. Otherwise, this paper contains high quality data and is well written.

We thank the reviewer for their contribution to the peer review process.

Minor comments:

l. 58: The ^{176}Lu – ^{177}Hf decay >> Th e ^{176}Lu – ^{176}Hf decay.

Fixed.

l. 71–118: It should be acknowledged in this section that some studies previously highlighted the fundamental Hf isotopic shift at ~3.8 Ga in the Hadean–Archean detrital zircon record (e.g., Bell et al., 2014; Bauer et al., 2020). Although these studies are included in the reference list, the main text gives an impression that the newly obtained data in this study reveal the isotopic shift for the first time, which is not the case.

References to Bell et al. and Bauer et al. are included in this section as requested.

Reviewer #2 (Remarks to the Author):

‘Crustal rejuvenation stabilised Earth’s first cratons’ by J. Mulder and colleagues

Upon reading this manuscript, I would reiterate my comments made on a previous version: it is clearly written, well illustrated, scientifically sound, and I believe would be of interest to the journal readership. The link between crustal stabilisation and formation of melt-depleted lithospheric mantle has been better explained and justified, and there is now greater attention to the geodynamic drivers, considering recent and relevant thermochemical/mechanical models. I am fully supportive of the publication of this manuscript and do not see how it could be easily improved within the scope of the study and the space restrictions.

Best regards, Tony Kemp, April 2021.

We thank the reviewer for their contribution to the peer review process.

A few minor points.

Line 26. A typo in the spelling of buoyant. Also in the caption to figure 4 and on the figure itself.

Fixed.

Line 55. I don't think ref. 55 is the most appropriate reference here. In my view, it is better to cite the seminal earlier papers that conceived this process – such as Barker and Arth (1976) *Geology*, or Rapp and Watson (1995) *J. Petrol.*

Reference to Rapp and Watson has been added.

Line 191. Here it would be appropriate to cite Petersson et al 2020 (*Chemical Geology*), who established a CHUR-like zircon Hf isotopic composition for the oldest gneissic rocks of the Pilbara Craton. This study also provides support for the points made in lines 61-63 and 141-143.

Reference to Petersson et al. has been added.